# Clifford Group Equivariant Simplicial Message Passing Networks

**Cong Liu**[12,*] **David Ruhe**[123,*]**, Floor Eijkelboom**[14]**, Patrick Forré**[12]
AMLab, University of Amsterdam
{c.liu4,d.ruhe,f.eijkelboom,p.d.forre}@uva.nl

## Abstract

We introduce Clifford Group Equivariant Simplicial Message Passing Networks, a method for steerable $E(n)$-equivariant message passing on simplicial complexes. Our method integrates the expressivity of Clifford group-equivariant layers with simplicial message passing, which is topologically more intricate than regular graph message passing. Clifford algebras include higher-order objects such as bivectors and trivectors, which express geometric features (e.g., areas, volumes) derived from vectors. Using this knowledge, we represent simplex features through geometric products of their vertices. To achieve efficient simplicial message passing, we share the parameters of the message network across different dimensions. Additionally, we restrict the final message to an aggregation of the incoming messages from different dimensions, leading to what we term *shared* simplicial message passing. Experimental results show that our method is able to outperform both equivariant and simplicial graph neural networks on a variety of geometric tasks. Our implementation is available on GitHub.

## 1 Introduction

Graph Neural Networks (GNNs) have established themselves as effective tools for learning representations of relational data from a variety of domains, including social networks (Fan et al., 2019), bioinformatics (Zitnik & Leskovec, 2017), and physics (Battaglia et al., 2016). Recently, there has been a surge of interest in the theoretical foundations of GNNs, with a particular focus on their expressive power (Geerts & Reutter, 2022). Standard message passing leverages the sparsity of the underlying graph by exchanging 'messages' between nodes only when they are adjacent. As such, nodes with local structures that are alike will learn similar representations, restricting the expressivity of this framework. This limitation has been formalized by Xu et al. (2019); Morris et al. (2019), showing that Message Passing Neural Networks (MPNNs) are at most as expressive as the Weisfeiler-Lehman (1-WL) test at distinguishing unattributed graphs. As a consequence, such networks cannot identify graph structures such as triangles or their higher-dimensional equivalents (Chen et al., 2020). In response, Bodnar et al. (2021) developed simplicial message-passing networks operating on simplicial complexes, resulting in both theoretical and empirical improvements regarding expressive power. Further, their contributions illuminate mathematical intersections with algebraic and differential topology, as well as geometry (Ghrist, 2014).

In the context of geometric graphs, the focus lies on graphs that are embedded in a geometry, such as a metric space or a manifold. These data points are often accompanied by geometric features, like positions or velocities. Such quantities transform predictably, though nontrivially, under rigid operations like rotations, reflections, or translations. $E(n)$ Equivariant Graph Neural Networks (EGNNs) (Satorras et al., 2021) are designed to respect these symmetries by either exhibiting an equivariance or invariance characteristic tailored to the specific task. This field has witnessed continual advancements (Huang et al., 2022; Thölke & Fabritiis, 2022; Finzi et al., 2020; Brandstetter et al., 2022; Batzner et al., 2022), with one of the latest being the introduction of Clifford Group Equivariant Neural Networks (CGENNs): a neural network architecture operating on the Clifford (or *geometric*) algebra (Ruhe et al., 2023a; Brehmer et al., 2023). While these methods reap the benefits of geometric equivariance, they are still limited to the expressive power of traditional message-passing.

---

*Contributed equally. [1]AMLab. [2]AI4Science Lab. [3]Anton Pannekoek Institute. [4]UvA-Bosch Delta Lab.

Combining the merits of both simplicial message passing and geometric equivariance, Eijkelboom et al. (2023) developed a method based on EGNNs, called $E(n)$ Equivariant Message Passing Simplicial Networks (EMPSNs). However, we can identify two limitations of this method. First, the higher-dimensional simplices are initialized using *manually* calculated geometric information. Second, EGNN is an architecture that falls under the umbrella of *scalarization* methods (Han et al., 2022), which operate mainly with invariant features and update vector features only through scaling. In this work, we go a step further and develop steerable Clifford algebra-based simplicial message-passing networks: Clifford Group Equivariant Simplicial Message Passing Networks (CSMPNs).

The Clifford algebra contains higher-order elements like bivectors and trivectors. These objects are computed through vector composition and can represent geometric quantities like areas and volumes. Analogously, simplices are also fully defined by their constituent vertices. The geometric product and the Clifford algebra are well-defined for any inner product space of any dimension. This versatility makes the geometric product a suitable option for computing simplex features and enables embedding simplices with geometric features across various spaces and dimensions.

As such, we initialize the higher-order simplices using geometric products (as well as linear combinations) of their vertices. The network then refines these simplices by passing messages between simplices of different order. To do so efficiently, unlike EMPSN, which uses a separate model for each type of communication, we share the parameters of the message passing function across various simplex orders, conditioning it on the dimensionalities of the source and target simplices. Additionally, we restrict the final message to an aggregation of the incoming simplicial messages, enabling the use of modern parallelized graph reduction operations in the simplicial setting. We call the resulting algorithm *shared simplicial message passing*. Equivariance is achieved by restricting all message and update neural networks to be Clifford group equivariant networks (Ruhe et al., 2023a). Experimental results show that our method can outperform both equivariant and simplicial graph neural networks in geometric tasks spanning multiple domains and dimensionalities, including a convex hulls volume prediction task, human walking motion prediction, molecular motion prediction, and NBA player trajectory prediction.

## 2 BACKGROUND

We provide a brief introduction to Clifford group equivariant neural networks and simplicial complexes. For a review on equivariant message passing networks, consider Appendix B.

### 2.1 CLIFFORD GROUP EQUIVARIANT NEURAL NETWORKS

We start by introducing the *Clifford algebra*, also known as the *geometric algebra*, which is a powerful mathematical object with applications in various areas of science and engineering. For a full development, we refer the reader to Ruhe et al. (2023a). While the theory has been generally developed, we restrict our attention to the case where the underlying vector space is $\mathbb{R}^d$. The Clifford algebra $\mathrm{Cl}(\mathbb{R}^d, q)$ is the "biggest" unitary, associative, non-commutative algebra generated by $\mathbb{R}^d$ such that for all $v \in \mathbb{R}^d$, $v^2 = q(v)$, where $q : \mathbb{R}^d \to \mathbb{R}$ is a quadratic form. In other words, vectors square to scalars. $q$ has an associated bilinear form $b : \mathbb{R}^d \times \mathbb{R}^d \to \mathbb{R}$. The algebra's elements, generally called *multivectors*, are linear combinations of non-commutative products of vectors while respecting the quadratic form: $x \in \mathrm{Cl}(\mathbb{R}^d, q)$, $x = \sum_{i \in I} c_i \cdot v_{i,1} \ldots v_{i,k}$. Here, the index set $I$ is finite, $c_i \in \mathbb{R}$ are scalars, and $v_{i,j} \in \mathbb{R}^d$ are vectors. Clifford multiplication is referred to as the *geometric product*. The Clifford algebra forms a graded vector space, with

$$\mathrm{Cl}(\mathbb{R}^d, q) = \bigoplus_{k=0}^d \mathrm{Cl}^{(k)}(\mathbb{R}^d, q), \qquad \dim \mathrm{Cl}^{(k)}(\mathbb{R}^d, q) = \binom{d}{k}, \qquad (1)$$

where and $\dim \mathrm{Cl}(\mathbb{R}^d, q) = 2^d$. These subspaces are called *grades*, where elements of $k = 0$ are called *scalars*, elements of $k = 1$ are called *vectors*, elements of $k = 2$ are called *bivectors*, and so on. A multivector $x \in \mathrm{Cl}(\mathbb{R}^d, q)$ can thus be written as a sum of its grades: $x = \sum_{k=0}^d x^{(k)}$, where $x^{(k)} \in \mathrm{Cl}^{(k)}(\mathbb{R}^d, q)$. The operation $(\cdot)^{(k)} : \mathrm{Cl}(\mathbb{R}^d, q) \to \mathrm{Cl}^{(k)}(\mathbb{R}^d, q)$ is called *grade projection*, which selects the grade-$k$ component of $x$. Beyond scalars and vectors, which only express magnitude and direction, respectively, bivectors express oriented area, trivectors express oriented volume, and so forth. Note that we really have the identifications: $\mathrm{Cl}^{(0)}(\mathbb{R}^d, q) = \mathbb{R}$ and $\mathrm{Cl}^{(1)}(\mathbb{R}^d, q) = \mathbb{R}^d$.

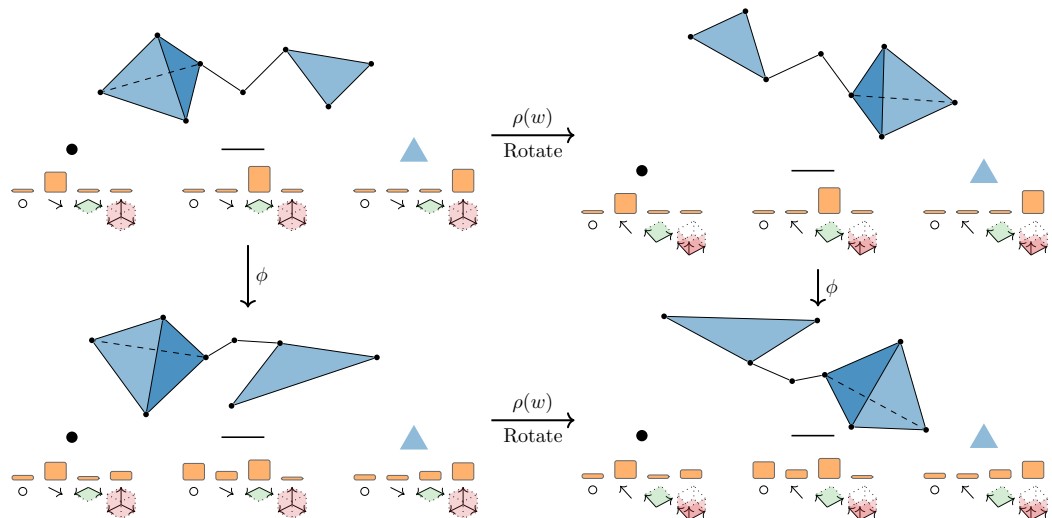

Figure 1: Illustration of our proposed architecture. Top left: a set of vertices (and edges) is lifted to a simplicial complex. We highlight three simplex types: vertices (0-simplices, ● ), edges (1-simplices, ── ), and triangles (2-simplices, ▲ ). In this case, the vertex feature is vector-valued and embedded as the grade 1 part of a Clifford algebra element: a *multivector*. In three dimensions, a multivector has scalar (○), vector (↗), bivector (⬡) and trivector (⬢) components. Higher-order simplices are initialized using the geometric product of their constituent vertices. As such, edges in the top left visualization are bivector-valued, and triangles are trivector-valued. The simplicial message-passing framework, denoted by $\phi$, refines the multivector-valued simplices, as portrayed in the bottom-left, by passing messages between simplices of different order. Crucially, $\phi$ maintains *equivariance* to the Clifford group's orthogonal action $\rho(w)$, representing a rotation here. In doing so, our method is ensured to respect the geometric symmetries of the input data.

Ruhe et al. (2023a) then define the *Clifford group* $\Gamma(\mathbb{R}^d, q)$. Its elements are invertible homogeneous[1] multivectors of $\mathrm{Cl}(\mathbb{R}^d, q)$ that preserve vectors under the group action $\rho(w) : \mathrm{Cl}(\mathbb{R}^d, q) \to \mathrm{Cl}(\mathbb{R}^d, q)$, called the *twisted conjugation*. That is, if $x \in \mathrm{Cl}^{(1)}(\mathbb{R}^d, q) = \mathbb{R}^d$, then $\rho(w)(x) \in \mathrm{Cl}^{(1)}(\mathbb{R}^d, q) = \mathbb{R}^d$. It is further shown that all $\rho(w)$ preserve the quadratic form $q$. Therefore, each $\rho(w)$ defines an *orthogonal* automorphism. Let $F \in \mathbb{R}[T_1, \ldots, T_\ell]$ be a polynomial (non-commutative, where multiplication is performed by the geometric product) in $\ell$ variables, $w \in \Gamma(\mathbb{R}^d, q)$, and $x_1, \ldots, x_\ell \in \mathrm{Cl}(\mathbb{R}^d, q)$. We then have the following equivariance properties (Ruhe et al., 2023a).

**Theorem 2.1** (All polynomials are Clifford group equivariant). *Let $F \in \mathbb{R}[T_1, \ldots, T_\ell]$ be a polynomial in $\ell$ variables with coefficients in $\mathbb{R}$, $w \in \Gamma(\mathbb{R}^d, q)$. Further, consider $\ell$ elements $x_1, \ldots, x_\ell \in \mathrm{Cl}(\mathbb{R}^d, q)$. Then we have the following equivariance property:*

$$\rho(w)\left(F(x_1, \ldots, x_\ell)\right) = F(\rho(w)(x_1), \ldots, \rho(w)(x_\ell)). \tag{2}$$

**Theorem 2.2** (All grade projections are Clifford group equivariant). *For $w \in \Gamma(\mathbb{R}^d, q)$, $x \in \mathrm{Cl}(\mathbb{R}^d, q)$ and $k = 0, \ldots, d$ we have the following equivariance property:*

$$\rho(w)(x^{(k)}) = \left(\rho(w)(x)\right)^{(k)}. \tag{3}$$

*In particular, for $x \in \mathrm{Cl}^{(k)}(\mathbb{R}^d, q)$ we also have $\rho(w)(x) \in \mathrm{Cl}^{(k)}(\mathbb{R}^d, q)$.*

That is, polynomials in multivectors, as well as their grade projections, are Clifford group equivariant, and therefore equivariant to the orthogonal group. It is worth mentioning that $\rho(w)$ not only preserves vectors, but is generally grade-preserving.

Using these two properties, Ruhe et al. (2023a) then introduce several linear, bilinear (through the geometric product), and nonlinear equivariant neural network layers. Composing these layers, we obtain a Clifford group equivariant neural network.

---

[1]We mean homogeneous with respect to *parity*. That is, multivectors that only have nonzero even grades ($k = 0 \mod 2$) or nonzero odd grades.

Several other works that incorporate geometric algebra in deep learning are (Melnyk et al., 2021; Spellings, 2021; Brandstetter et al., 2023; Ruhe et al., 2023b). Finally, Brehmer et al. (2023) introduce the geometric algebra transformer, a method that uses the *projective* geometric algebra (Gunn, 2016) in combination with the transformer (Vaswani et al., 2017) architecture.

## 2.2 SIMPLICIAL COMPLEXES

**Definition 2.3** (Simplicial Complex). *Let $V$ be a finite set. An abstract simplicial complex $K$ is a subset of the power set $2^V$ that satisfies:*

1. *$\forall v \in V : \{v\} \in K$;*

2. *$\forall \sigma \in K : \forall \tau \subseteq \sigma, \tau \neq \emptyset : \tau \in K$.*

In words, it is a space formed by a collection of subsets $\sigma \subseteq V$, called *simplices*, which always including all the singletons, such that if $\sigma \in K$ and $\tau \subseteq \sigma$, then $\tau \in K$ as well. In other words, $K$ is closed under taking subsets. For example, if a triangle (dimension 2 simplex) is in $K$, then all its edges and vertices (dimensions 1 and 0, respectively) are also in $K$. We also have $\dim \sigma := |\sigma| - 1$ and $\dim K := \max\{\dim \sigma \mid \sigma \in K\}$. Note that the 1-*skeleton* of a simplicial complex is a graph consisting of only vertices and edges, showing that simplicial complexes are a natural generalization of graphs. The act of creating a simplicial complex from a lower dimensional ones, like a vertex set, is called a *lifting* transformation. By constructing a rich adjacency structure on the simplicial complex, we can model interactions not only between vertices, but also between higher-dimensional simplices. E.g. a triangle in a simplicial complex can be thought of as a *meta-vertex* that represents the interaction between its three vertices. Bodnar et al. (2021) introduce message passing networks on simplicial complexes lifted from regular graphs. Exploring the connectivity within a simplicial complex, they identify the following adjacency types.

**Definition 2.4** (Simplicial Adjacencies). *Let us define the boundary relation $\tau \prec \sigma := \tau \subset \sigma \wedge \dim \tau = \dim \sigma - 1$. We define the following adjacency structures of a simplicial complex $K$:*

1. *Boundary adjacencies $B(\sigma) := \{\tau \in K \mid \tau \prec \sigma\}$;*

2. *Coboundary adjacencies $C(\sigma) := \{\tau \in K \mid \tau \succ \sigma\}$;*

3. *Lower adjacencies $N_\downarrow(\sigma) := \{\tau \in K \mid \exists \delta \in K : \delta \prec \tau, \delta \prec \sigma\}$;*

4. *Upper adjacencies $N_\uparrow(\sigma) := \{\tau \in K \mid \exists \delta \in K : \delta \succ \tau, \delta \succ \sigma\}$.*

For example, if $\sigma$ is a triangle, then $B(\sigma)$ is the set of edges that make up the triangle. Lower adjacencies are simplices of the same dimensionality but with a common lower boundary, while upper adjacencies share a common upper boundary. Note that regular message passing uses only the upper adjacencies of nodes. Simplicial message passing networks satisfy the following theorems.

**Theorem 2.5** (Bodnar et al. (2021)). *Simplicial Weisfeiler-Lehman (SWL) with a clique complex lifting is strictly more powerful than Weisfeiler-Lehman (WL), and is not less powerful than 3-WL.*

**Theorem 2.6** (Bodnar et al. (2021)). *Message Passing Simplicial Networks (MPSNs) with sufficient layers and injective neighborhood aggregators are as powerful as SWL. Moreover, MPSNs are strictly more powerful than WL.*

The Weisfeiler-Lehman algorithm (Leman & Weisfeiler, 1968) is a well-known graph isomorphism test. Here, a *clique complex lift* turns every clique in a graph into a simplex in a simplicial complex. We depict two isomorphic geometric graphs in Appendix C.

## 3 CLIFFORD GROUP EQUIVARIANT SIMPLICIAL MESSAGE PASSING NETWORKS

### 3.1 CREATING GEOMETRIC SIMPLICIAL COMPLEXES

Starting with a finite set $V$, potentially part of a (geometric) graph $G = (V, E)$, several ways exist to construct a simplicial complex $K$. For instance, we can form a simplicial complex by creating a

simplex for every possible subset of $V$. That is, we include all edges, triangles, tetrahedra, and so on. However, the resulting simplicial complex would have $\binom{|V|}{n+1}$ number of $n$-simplices. One can easily see that this number can quickly become prohibitively big.

As such, various methods exist to *lift* a set of points to a simplicial complex in a more tractable way. Examples are *Vietoris-Rips*, *Čech*, manual and algorithmic lifts. Furthermore, if we have access to a graph structure, we can use a *clique lift*. For a more detailed discussion of these, consider Appendix D. To create a Vietoris-Rips lift, we consider a node position $x^v \in X$ attached to $V$, together with a distance function $d : X \times X \to \mathbb{R}$. Then, the Vietoris-Rips complex $\text{Vietoris-Rips}(V, d, \epsilon)$ is the simplicial complex containing all simplices whose vertices are $\epsilon$-close for some $\epsilon > 0$ (see Figure 5). The manual lift defines a simplicial complex by hand. For example, we can define the shape of a polyhedral object by its vertices, edges, and faces, making a simplicial complex also known as a *mesh*. Another alternative is to let expert knowledge guide us. Take, for example, the molecule of water $H_2O$. The bond angle between the two hydrogen atoms, governed by triangular interaction involving all three atoms, is crucial for the molecule's properties. Finally, recent work has been done on algorithmically constructing simplicial complexes from data, for example, through the *mapper* procedure (Hajij et al., 2018). In this work, we use the Vietoris-Rips and manual lifts in our experiments. We typically cap the maximal simplex dimension to 2 for efficiency reasons.

### 3.2 Embedding Simplicial Data in the Clifford Algebra

In the following, we elaborate on how we embed the usual scalar and vector features of each node $v \in V$ in the Clifford algebra, as well as how we create simplex features.

First, we have node features $\forall v \in V : h^v \in \mathbb{R}^k \oplus \left(\mathbb{R}^d\right)^l$. Here, let $h_1^v, \ldots, h_k^v \in \mathbb{R}$ denote *scalar* features, or *invariants*. Consider, for example, a particle's mass or charge in a physics setting. They transform trivially under the Clifford group action. On the other hand, let $h_{k+1}^v, \ldots, h_l^v \in \mathbb{R}^d$ be *vector* features, such as position, velocity, and acceleration. They transform nontrivially under the Clifford group action. One of these is usually the position of the vertex, which (e.g., through distances) is used to construct a simplicial complex. Recall that the Clifford subspace $\text{Cl}^{(0)}(\mathbb{R}^d, q) = \mathbb{R}$ is the subspace of all scalars, and therefore we can embed our scalar features in this subspace. $\text{Cl}^{(1)}(\mathbb{R}^d, q) = \mathbb{R}^d$ is the vector subspace, and therefore we can embed our vector features here. Further, if we have access to higher-order features such as bivectors, we can embed them in the other subspaces of the Clifford algebra. As such, after embedding, we now denote $\forall v \in V : h^v \in \text{Cl}(\mathbb{R}^d, q)^m$, where $m = k + l$.

We now consider a simplicial complex $K$ lifted from $V$. Our goal is, analogously to the node features, to obtain a Clifford feature $h^\sigma$ for each simplex $\sigma \in K$. For the singletons $\{v_i\} \in K$, we can directly put $h^{\{v_i\}} := h^{v_i}$. For the edges $\{v_i, v_j\} \in K$, we can put $h^{\{v_i, v_j\}} := h^{v_i} h^{v_j}$, denoting the geometric product of the two Clifford features. This process extends to triangles and higher-dimensional simplices, multiplying Clifford features of all vertices with geometric product. In Figure 1, we depict this embedding, illustrating how edge and triangle simplices relate to bivector- and trivector-valued features. Since there are multiple ways to embed simplices, we *learn* the embedding through Clifford group-equivariant layers, i.e., we take the Clifford simplicial features as input to learnable Clifford group-equivariant layers and use the outputs as the learnable Clifford simplicial features. These can be decomposed into parameterized linear combinations as well as parameterized geometric products, resulting in analogous embeddings to the ones described above, but including learnable parameters. To ensure permutation-invariant embeddings, one can aggregate permutations of geometric products. A more intricate approach involves passing messages between the vertices of a $k$-simplex. The aggregated readout is then used as the initialized feature for the $k$-simplex.

Eijkelboom et al. (2023) take a similar approach as here, but they *manually* have to define the simplicial embeddings for all simplices. Specifically, distances, volumes, and angles between simplices are calculated. All such quantities can be expressed using linear combinations and geometric products (Doran & Lasenby, 2003), which are computed by Clifford group equivariant layers. Moreover, Eijkelboom et al. (2023) can only embed geometrically *covariant* information through relative positions, which are computed *additively*. In contrast, the use of geometric products and the fact that any polynomial in multivectors is Clifford group equivariant, we can also embed covariant information

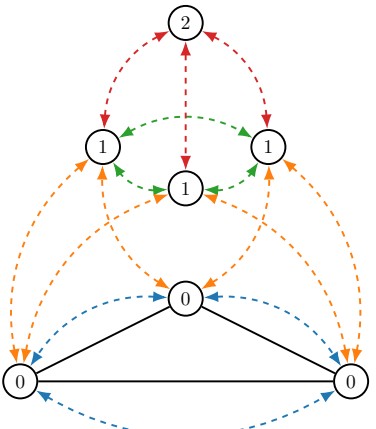 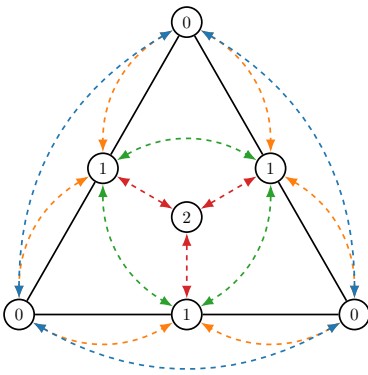

Figure 2: Left: we show how a simple graph (three fully-connected nodes) is lifted to a simplicial complex. Using simplicial message passing, we allow communication between objects of different dimensions. That is, between vertices $(0 \leftrightarrow 0)$ ◄--►, nodes and edges $(0 \rightarrow 1$ and $1 \rightarrow 0)$ ◄--►, edges $(1 \leftrightarrow 1)$ ◄--►, and between edges and triangles $(1 \rightarrow 2$ and $2 \rightarrow 1)$ ◄--►. Right: same as left, but a top-down view. It illustrates the hypergraph associated with the complex with several *meta-vertices* representing the simplices of various dimensionality. Instead of running message passing separately for all different communication types, we share the parameters of a single neural network operating on the extended graph. By conditioning on the message type, it is still able to leverage the simplicial complex.

extracted from three or more node positions *multiplicatively*. Since these networks generalize to inner-product spaces of any dimension, these notions also hold for Clifford algebras over higher-dimensional spaces.

### 3.3 EQUIVARIANT SHARED SIMPLICIAL MESSAGE PASSING

For all $\sigma \in K$, we now have a Clifford feature $h^\sigma \in \mathrm{Cl}(\mathbb{R}^d, q)^m$. We propose two techniques that enable efficient (equivariant) message passing on simplicial complexes. In doing so, we require access to a parameterized message function $\phi^m$ and an update function $\phi^h$. First, the message $m^\sigma$ will be an equivariant aggregation of all information (processed by a neural network) from several adjacencies of different dimensions as defined in Definition 2.4 [2]. In contrast to, e.g., Eijkelboom et al. (2023); Bodnar et al. (2021), who iteratively run message passing for different adjacency types, we can leverage existing parallel implementations of classical message passing. In other words, we consider the adjacency matrix of the simplicial complex's corresponding *hypergraph*, where we have several *meta-vertices* that represent the different types of simplices. This corresponds to considering the $0$-simplices of the *barycentric subdivision* of the simplicial complex, which is a common way for refining simplicial complexes (Ghrist, 2014). This idea is visualized in Figure 2.

Secondly, instead of considering a different parameterization for each type of communication, we define a single message function $\phi^m$ that can handle all types of communication. However, by conditioning $\phi^m$ on the type of message, it can still leverage the simplicial complex. In doing so, we efficiently share parameters between different types of communication, which is in contrast with previous methods that defined a different neural network for each type of communication.

---

**Algorithm 1** Shared Simplicial Message Passing

**Require:** $K, \forall \sigma \in K : h^\sigma, \phi^m, \phi^h$
**Repeat:**
$$m^\sigma \leftarrow \underset{\substack{\tau \in B(\sigma) \\ \tau \in C(\sigma) \\ \tau \in N_\uparrow(\sigma) \\ \tau \in N_\downarrow(\sigma)}}{\mathrm{Agg}} \phi^m(h^\sigma, h^\tau, \dim \sigma, \dim \tau)$$
$$h^\sigma \leftarrow \phi^h\left(h^\sigma, m^\sigma, \dim \sigma\right)$$

---

[2]Intriguingly, Bodnar et al. (2021) prove that only the boundary and upper adjacencies are required for full expressivity. However, we only use the boundary, coboundary, and upper adjacencies to keep consistent with Eijkelboom et al. (2023).

|  | MSE ($\downarrow$) |
|---|---|
| MPNN (Gilmer et al., 2017) | 0.212 |
| GVP-GNN (Jing et al., 2021) | 0.097 |
| VN (Deng et al., 2021) | 0.046 |
| EGNN (Satorras et al., 2021) | 0.011 |
| CGENN (Ruhe et al., 2023a) | 0.013 |
| EMPSN (Eijkelboom et al., 2023) | 0.007 |
| **CSMPN** | **0.002** |

Table 1: MSE ($\downarrow$) of the tested models on the convex hulls experiment.

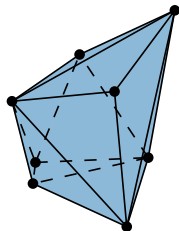

Figure 3: In the convex hulls experiment, the task is to estimate the volume of the convex hull of eight *five-dimensional* random points. Here, we display a three-dimensional example, which is easier to visualize.

To make the overall method equivariant, we utilize Clifford group equivariant neural networks from Ruhe et al. (2023a). Then, as long as the simplicial embedding, the aggregation operation (e.g., a summation), and $\phi^m$ and $\phi^h$ are equivariant, the overall method is Clifford group equivariant (see Appendix B). This then makes it equivariant to rotations, reflections, and other orthogonal transformations in any dimension. The algorithm is summarized in Algorithm 1. Note that it generalizes typical message passing, which only considers messages from the upper adjacencies between 0-simplices, i.e. nodes. For a quick overview of how this compares to typical message passing and simplicial message passing, consider Appendix F.

## 4 EXPERIMENTS

We selected a set of geometric experiments that involve different types of data from several domains and include both invariant and equivariant predictions. Currently, the QM9 (Ramakrishnan et al., 2014) and MD17 (Chmiela et al., 2017) datasets are highly popular benchmarks for equivariant graph neural networks. We found, however, that the state-of-the-art models incorporate a lot of domain knowledge, which is beyond the scope of the current work. Note that we ensure a fair comparison by maintaining a similar scale of parameters between CSMPN and the baseline models across all experiments. More experimental details than presented here can be found in Appendix E.

### 4.1 5D CONVEX HULLS

We run this experiment based on the convex hull volumetric experiment of Ruhe et al. (2023a). We consider a *five-dimensional* space, where we sample eight points from a standard normal distribution. The task is to estimate the volume of the convex hull of these points. We give an example of the three-dimensional case in Figure 3. Note that this is an E(5)-invariant task. For the simplicial networks (EMPSN (Eijkelboom et al., 2023) and ours), we use the representation of the hull as a set of 4-simplices, and extract all their 1- and 2-simplices. We then pass messages between these as well as the 0-simplices (the points). As shown in Table 1, CSMPN outperforms the other models, showing that in this setting the simplicial structure is a significantly improved way of presenting the data to the network.

### 4.2 CMU MOTION CAPTURE

In this experiment, we evaluate our models on the CMU Human Motion Capture dataset (Gross & Shi, 2001). We demonstrate that CSMPN exhibits greater performance in motion prediction compared to equivariant architectures reliant on regular graphs. In agreement with, e.g., Huang et al. (2022), we use the 35th human subject of the dataset. Each graph in the dataset contains 31 equally connected nodes, with each representing a specific position on the human body during walking. The objective is to use the node positions of a random frame to predict the node positions after 30 timesteps. We *manually* lift the graph to a simplicial complex. Specifically, we connect elbow joint nodes with shoulder and palm nodes to create edges and triangles. Similarly, hip, knee, and heel nodes are linked, forming another set of edges and triangles.

| Method | MSE (↓) |
|---|---|
| Radial Field (Köhler et al., 2020) | 197.0 |
| TFN (Thomas et al., 2018) | 66.9 |
| SE(3)-Tr (Fuchs et al., 2020) | 60.9 |
| GNN (Gilmer et al., 2017) | 67.3 |
| EGNN (200K) (Satorras et al., 2021) | 31.7 |
| GMN (200K) (Huang et al., 2022) | 17.7 |
| EMPSN (200K) | 15.1 |
| CGENN (200K) | 9.41 |
| CSMPN (200K) | **7.55** |

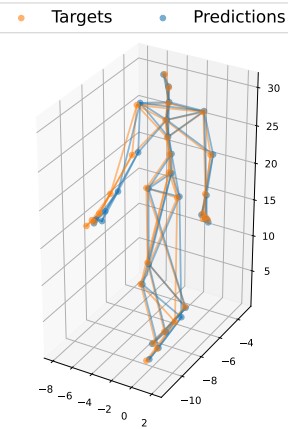

Table 2: Left: MSE ($10^{-2}$) of the tested models on the CMU motion capture dataset. Right: Depiction (not cherry-picked) of an instance (the ground-truth target positions) vs. a CSMPN prediction.

| | Aspirin | Benzene | Ethanol | Malonaldehyde |
|---|---|---|---|---|
| Radial Field (Köhler et al., 2020) | 17.98 / 26.20 | 7.73 / 12.47 | 8.10 / 10.61 | 16.53 / 25.10 |
| TFN (Thomas et al., 2018) | 15.02 / 21.35 | 7.55 / 12.30 | 8.05 / 10.57 | 15.21 / 24.32 |
| SE(3)-Tr (Fuchs et al., 2020) | 15.70 / 22.39 | 7.62 / 12.50 | 8.05 / 10.86 | 15.44 / 24.47 |
| EGNN (Satorras et al., 2021) | 14.61 / 20.65 | 7.50 / 12.16 | 8.01 / 10.22 | 15.21 / 24.00 |
| S-LSTM (Alahi et al., 2016) | 13.12 / 18.14 | 3.06 / 3.52 | 7.23 / 9.85 | 11.93 / 18.43 |
| NRI (Kipf et al., 2018) | 12.60 / 18.50 | 1.89 / 2.58 | 6.69 / 8.78 | 12.79 / 19.86 |
| NMMP (Hu et al., 2020) | 10.41 / 14.67 | 2.21 / 3.33 | 6.17 / 7.86 | 9.50 / 14.89 |
| GroupNet (Xu et al., 2022) | 10.62 / 14.00 | 2.02 / 2.95 | 6.00 / 7.88 | 7.99 / 12.49 |
| GMN-L (Huang et al., 2022) | 9.76 / - | 48.12 / - | 4.83 / - | 13.11 / - |
| EqMotion (300K) (Xu et al., 2023) | 5.95 / 8.38 | 1.18 / 1.73 | 5.05 / 7.02 | 5.85 / 9.02 |
| EMPSN (300K) | 9.53 / 12.63 | **1.03 / 1.12** | 8.80 / 9.76 | 7.83 / 10.85 |
| CGENN (300K) | **3.70 / 5.63** | **1.03** / 1.59 | 4.53 / 6.35 | 4.20 / 6.55 |
| CSMPN (300K) | 3.82 / 5.75 | **1.03** / 1.60 | **4.44 / 6.30** | **3.88 / 5.94** |

Table 3: ADE / FDE ($10^{-2}$) (↓) of the tested models on the MD17 atomic motion dataset.

We incorporated all baselines from Huang et al. (2022) and added EMPSN and CGENN ourselves. From Table 2, we note that EMPSN, the simplicial version of EGNN, already enjoys a significant boost from simplicial message passing. On top of this, Clifford layers further improve the performance, which CSMPN then again surpasses. We ensured that model sizes, in terms of the number of parameters, are comparable.

### 4.3 MD17 Atomic Motion Prediction

We turn to the molecular domain with the MD17 dataset Chmiela et al. (2017). However, we do not use the standard task of predicting the energy of a molecule, but instead we predict the motion of the atoms. In accordance with Han et al. (2022), we select four molecules: aspirin, benzene, ethanol, and malonaldehyde. The aim is to assess how well CSMPN can model molecular dynamics by directly predicting atom positions in future time steps. We take the starting positions of the heavy atoms from ten separate time frames for each molecule and predict their positions in the next ten frames. For aspirin, we use $k$-nearest neighbors with $k = 3$ to construct the regular graphs from the atom positions. The other molecules are fully connected. Since we have access to this graph structure, we use a clique complex lift to form the simplicial complex.

The Average Displacement Error / Final Displacement Error (ADE / FDE) of the methods are displayed in Table 3. Here, ADE is the RMSE of the location predictions averaged over the number of

|  | Attack | Defense |
|---|---|---|
| STGAT (Huang et al., 2019) | 9.94 / 15.80 | 7.26 / 11.28 |
| Social-Ways (Amirian et al., 2019) | 9.91 / 15.19 | 7.31 / 10.21 |
| Weak-Supervision (Zhan et al., 2019) | 9.47 / 16.98 | 7.05 / 10.56 |
| DAG-Net (200K) (Monti et al., 2020) | 8.98 / 14.08 | 6.87 / 9.76 |
| CGENN (200K) | 9.17 / 14.51 | 6.64 / 9.42 |
| CSMPN (200K) | **8.88 / 14.06** | **6.44 / 9.22** |

Table 4: ADE / FDE ($\downarrow$) of the tested models on the VUSport NBA player trajectory dataset.

time-steps, and FDE the RMSE of the final time-step. We take all baselines from Xu et al. (2023) and included EMPSN and CGENN ourselves. Inspecting the results, we see that simplicial message passing yields a significant boost for EMPSN, the simplicial version of EGNN. Clifford layers then further improve the performance, the only exception being the case of benzene. We hypothesize that the restricted expressivity of EMPSN forms a good inductive bias here, since the molecule is rigid and planar. CSMPN achieves outstanding scores across the board, slightly surpassing clifford in the cases of ethanol and malonaldehyde. We ensured that the simplicial structure used in the comparison of EMPSN and CSMPN is equal, and that the number of parameters is comparable. Similarly, the architecture of the CGENN and CSMPN models is roughly identical.

## 4.4 NBA Players 2D Trajectory Prediction

Finally, we subject our CSMPN to testing on the STATS SportVU NBA Dataset (STATS Perform, 2023). This *two-dimensional* dataset contains the tracking positions of the NBA team players in regular seasons. Our preprocessing approach aligns with Monti et al. (2020), representing each player's position as a two-dimensional coordinate. Considering the distinct behaviors of players in offensive or defensive roles, we assess our model on these different contexts, utilizing ten observed time frames to predict the subsequent forty. Motion uncertainty and interactions between players make the task challenging. Each player's position is represented as a node in our graph, each node is connected to all the other nodes in the graph to make the regular graph fully-connected. We also include one fixed reference point in our graphs indicating the orientation of the basketball court. We create the simplicial complex using Vietoris-Rips with infinite $\epsilon$ with a maximum simplex dimension of 2. We compare our CSMPN with baselines provided by Monti et al. (2020) as well as CGENN and present the results in Table 4.

## 5 Conclusion

We presented Clifford Group Equivariant Simplicial Message Passing Networks (CSMPNs), a class of Clifford algebra-based neural networks that are $\mathrm{E}(n)$-equivariant and operate on simplicial complexes. Our method links the combination of vertices to form higher-dimensional simplices to the combination of vectors to form multivectors, combining the expressiveness of Clifford group equivariant neural networks with the topological intricacy of simplicial message passing. To do so efficiently, we reduce simplicial message passing to message passing on a *hypergraph*, but condition on the dimensionality of the source and target simplices. As such, we can share the parameters of the message passing functions across various simplex orders, thereby maintaining the topological structure of the complex. Experimental results showed that CSMPNs can outperform both equivariant and simplicial graph neural networks on a variety of tasks. In some cases, however, the performance of CSMPNs is comparable to that of the other methods. Future research might explore which scenarios specifically benefit from utilizing simplicial geometric message passing. While a limitation to method is the increased computational cost of simplicial message passing, we have already made significant strides by sharing parameters across simplex orders and are optimistic about future improvements in this direction.

ACKNOWLEDGMENTS

The author C.L. was financially supported by Health-Holland, Top Sector Life Sciences & Health (LSH-TKI), project number LSHM22023, which realizes a public-private partnership (PPP) between the University of Amsterdam and Janssen Vaccines & Prevention B.V. We thank SURF (www.surf.nl) for the support in using the National Supercomputer Snellius.

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

## A    REPRODUCBILITY & BROADER IMPACT

We will make the complete codebase utilized in our experiments publicly available, encompassing aspects like model architectures, data preprocessing, training configurations, hyperparameters, and evaluation methodologies. This open-source approach is aimed at ensuring straightforward reproducibility.

Enhancing current graph-based models carries the promise of advancing numerous scientific domains, including but not limited to computational biology, materials science, and computer vision. These advancements have the potential to catalyze discoveries and innovations that contribute to the betterment of society by deepening our comprehension of intricate, organized datasets.

## B    EQUIVARIANT MESSAGE PASSING NETWORKS

Symmetry is crucial in various branches of mathematics and science, such as geometry, physics, and chemistry, for understanding the underlying structures and properties of objects and systems. Groups are mathematical formalizations of symmetries, where each element of the group corresponds to a symmetry of the object, and the group operation combines these symmetries.

In geometric deep learning (Bronstein et al., 2021), equivariance refers to a property where the output of a model $\phi : X \to Y$ should transform predictably when a transformation from some symmetry group is applied to the input. Let $w$ be an element of an abstract group $G$. We have a representation $\rho_X(w)$ (e.g., a rotation matrix)[3] on a vector space $X$, such that $\rho_X(w) : X \to X$. If we have

$$\phi(\rho_X(w)(x)) = \rho_Y(w)(\phi(x)) \tag{4}$$

for all $x \in X$ and $w \in G$, then $\phi$ is called $G$-equivariant.

Equivariance is a desirable property of neural networks since it ensures that they respect certain symmetries present in the data. Instead of representations breaking down, equivariant neural networks ensure that their output transforms faithfully. This stability is crucial for neural networks to generalize well to unseen data. Moreover, equivariant models are more data-efficient, as not all possible symmetries need to be learned from data.

Message Passing Neural Networks (Gilmer et al., 2017) are neural algorithms for learning on graphs. We denote a graph $G = (V, E)$ to be a tuple of nodes and edges. A node $v \in V$ or edge $e = (v, w) \in E$ can have attached features $h^v$ or $h^{(v,w)}$, respectively. We then have the following update rules:

$$m_l^v := \text{Agg}_{w \in N(v)} \left( \phi^m(h_l^v, h_l^w, h^{(v,w)}) \right); \tag{5}$$

$$h_{l+1}^v := \phi^h(h_l^v, m_l^v), \tag{6}$$

where $N(v)$ denotes the neighbor set of $v$ and $h_0^v := h^v$. Here, $\phi^m$ and $\phi^h$ are neural networks, called the *message* and *update* functions, respectively. Further, Agg denotes a *permutation-invariant* aggregation function, e.g., a summation or average.

*Locality* is a physics principle that states that an object is influenced only by its immediate surroundings. Message passing allows a model to capture local information by symmetrically characterizing the relationships between a central node and its adjacent nodes. This mechanism enables the model to respect the inherent spatial hierarchies and dependencies present within the data. Further, sharing parameters of message and update networks across different nodes in graphs reduces parameters of networks but also provides consistency in learning structural relationships.

---

[3]Note that $\rho_X(w)$ does not necessarily have to be a linear function; it can also exist as a non-linear function within a different function space.

*Equivariant* message passing can be achieved by restricting the message and update functions, together with the aggregation, to be $G$-equivariant in all their arguments:

$$\rho(w)(m_l^v) = \rho(w)\left(\text{Agg}_{w \in N(v)}\left(\phi^m(h_l^v, h_l^w, h^{(v,w)})\right)\right) \tag{7}$$

$$= \text{Agg}_{w \in N(v)}\left(\rho(w)\left(\phi^m(h_l^v, h_l^w, h^{(v,w)})\right)\right) \tag{8}$$

$$= \text{Agg}_{w \in N(v)}\left(\phi^m(\rho(w)(h_l^v), \rho(w)(h_l^w), \rho(w)(h^{(v,w)}))\right). \tag{9}$$

Similarly, for the update equation.

$$\rho(w)(h_{l+1}^v) = \rho(w)\left(\phi^h(h_l^v, m_l^v)\right) \tag{10}$$

$$= \phi^h(\rho(w)(h_l^v), \rho(w)(m_l^v)). \tag{11}$$

Here, we suppressed the representation space on which $\rho(w)$ acts for brevity. By composing these layers, we obtain a $G$-equivariant neural network that ensures that all outputs are transformed predictably under a $G$-action applied to the input.

## C GEOMETRIC GRAPH ISOMORPHISM: EXAMPLE

.

In Figure 4 we display two non-isomorphic geometric graphs. That is, the node attributes may contain geometric information, e.g., their position in space. A regular graph neural network would not be able to distinguish such two graphs. That is, their representations would be equal. However, by lifting the graphs to simplicial complexes and passing messages between different types of simplices, we can distinguish

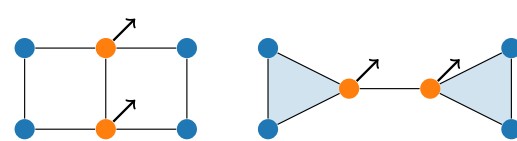

Figure 4: Two non-isomorphic geometric graphs.

between the two graphs. Furthermore, if we apply a geometric transformation to the geometric features, such as a rotation, the representations of the non-equivariant models would break down. For equivariant models, the representations would be transformed accordingly, respecting the fact that the chosen frame or reference is arbitrary.

## D CONSTRUCTING SIMPLICIAL COMPLEXES

Since data is typically not presented as graphs or point clouds rather than simplicial complexes, a choice needs to be made when constructing a simplicial complex based on the underlying graph. We outline four different approaches. We assume that our input consists of either geometric graphs $G = (V, E)$ such that for each $v \in V$ we have some coordinate $x \in X$, or point clouds $\{x_i\}_{i=1}^N$ of $N$ coordinates. Note that for the second case, there is no underlying graph structure present that can be leveraged to create the complex.

**Clique Lifts** If we have a geometric graph, a standard graph lift or *clique lift* can be used. Since a $n$-simplex is defined by $n+1$ points that are fully connected (e.g., a triangle or 2-simplex consists of three fully connected points), we can naturally associate to each clique of $n + 1$ points in the graph a $n$ simplex.

**Manual Lifts** In the situation where we do not have a graph or want to augment an existing graph, we sometimes can use domain knowledge to construct the simplicial complexes. That is, if we have some intuition about where adding simplices in the graph could be beneficial, we can create a simplicial complex by manually adding simplices to the data. Take, for example, the molecule of water $H_2O$. The bond angle between the two hydrogen atoms is crucial for the properties of water, and this interaction involves all three atoms. Modeling this requires a structure that goes beyond pairwise interactions.

Moreover, manual lift can be initiated with the Vietoris-Rips lift to form simplicial complexes. Subsequently, one can impose manually set thresholds for different dimensional simplices (like

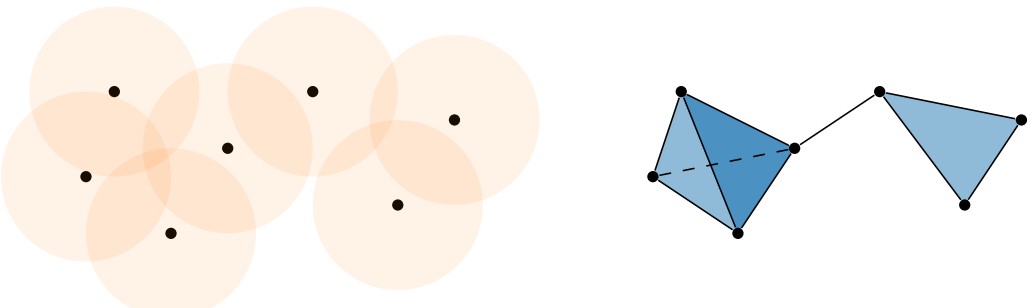

Figure 5: Left: a two-dimensional point cloud with $\epsilon$-balls around each point. Right: the corresponding Vietoris-Rips simplicial complex. In this case, it is equal to the Čech complex.

edge lengths for 1-simplices, areas of triangles for 2-simplices, and volumes of tetrahedra for 3-simplices). A group of (geometrically) close nodes form higher-order chemical structures, such as functional groups (think: aldehydes, ketones, nitrates, etc). E.g., if we want to predict some properties of (amino) acids, it could be useful to have a "simplex" correspond with a hydrophilic group s.a. carboxyl. Additionally, since molecular graphs often contain carbon rings, it's logical to break these rings into 2-dimensional or higher-order simplicial complexes to better represent the rings in a topological sense. Moreover, higher-order simplices can be manually constructed based on expert knowledge to capture the specific chemical and physical properties of the molecules. For instance, in biochemical structures like proteins or DNA, certain arrangements of atoms or residues have specific functional implications. By manually constructing higher-order simplices, one can incorporate these domain-specific insights into the model. This approach allows for the explicit representation of essential molecular structures such as hydrogen bonding patterns, aromatic systems, which are critical for understanding molecular behavior and interactions.

A similar approach is commonly leveraged in computer graphics, where objects are typically crafted and then approximated with meshes, creating 2-dimensional simplicial complex embedded in $\mathbb{R}^3$.

**Viertoris-Rips and Čech Complexes**     In the case of no graph or domain information available, we can use a *Viertoris-Rips* complex or a *Čech* complex. For this, suppose our point cloud is embedded in some metric space $(X, d)$ where $d : X \times X \to \mathbb{R}$ is some arbitrary distance function; e.g., the norm between vectors in $\mathbb{R}^n$. Both approaches construct $\epsilon$-balls around points to form a simplicial complex based on the geometry of the space. Formally, we have for $\epsilon > 0$

$$\text{Vietoris-Rips}(V, d, \epsilon) := \{\sigma \subset V \mid \forall v, w \in \sigma : d(x^v, x^w) \leq \epsilon\} \tag{12}$$

That is, the Vietoris-Rips complex $\text{Vietoris-Rips}(X, d, \epsilon)$ is the simplicial complex containing all simplices whose vertices are $\epsilon$-close. Similarly, we can construct *Cech complex* as

$$\text{Čech}(V, d, \epsilon) := \{\sigma \subset V \mid \bigcap_{v \in \sigma} B(v, \epsilon) \neq \emptyset\}, \tag{13}$$

where $B(v, \epsilon) := \{x \in X \mid d(x^v, x) \leq \epsilon\}$ is the closed ball of radius $\epsilon$ around $x^v$. See Figures 5 and 6 for an illustrative example of these two lifts.

Intuitively, if we consider the topology formed by the union of $\epsilon$-balls, the case can be made that Čech complexes more intuitively resemble the topology on the data since $\text{Čech}(V, d, \epsilon)$ is homotopy-equivalent to the topology formed by combining these $\epsilon$-balls. However, the runtime for constructing a Vietoris-Rips complex is significantly less in practice, as illustrated in Eijkelboom et al. (2023). Moreover, since it holds that for any $\epsilon > 0$ we have

$$\text{Čech}(V, d, \epsilon) \subset \text{Vietoris-Rips}(V, d, \epsilon) \subset \text{Čech}(V, d, 2\epsilon), \tag{14}$$

we know that if we can find some $\epsilon$ such that the data is well described by the respective Čech complexes, then so will it be by a Vietoris-Rips complex.

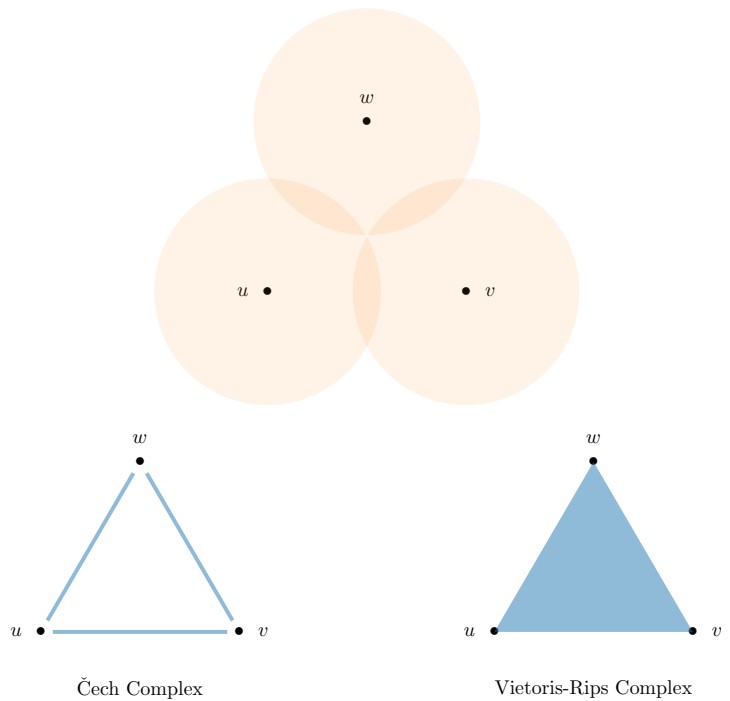

Figure 6: We show three nodes $u, v, w$, their respective $\epsilon$-balls, as well as the resulting Čech complex and Vietoris-Rips complex. Since the intersection of the balls is empty, the corresponding Čech complex does not contain triangle simplex, whereas the Vietoris-Rips complex does.

# E EXPERIMENTAL DETAILS

## E.1 5D CONVEX HULLS

The hulls are generated by randomly choosing eight nodes around the origin. We then use the convex hull functionality of *Scipy* (Virtanen et al. (2020)) to realize it. The task is to predict the volume of the convex hulls given just the positions of the nodes. Graph neural network baselines operate on a fully-connected graph created from the positions.

An $n$-dimensional convex hull can be represented by a set of $(n-1)$-simplices. In our case, $n = 5$. We then consider all sub-simplices of these $(n-1)$-simplices up to dimension 2 and pass messages between them.

The finalized dataset contains 16384 entries for training, validation, and test sets. To maintain objectivity in the comparison, it is ensured that all the baseline models and CSMPN have roughly equivalent number of parameters (200K). We use three simplicial message passing layers where the message and update functions are Clifford group-equivariant MLPs with 28 hidden features. Training of CSMPN is achieved through an Adam optimizer (Kingma & Ba, 2017) with a learning rate of $1 \times 10^{-3}$. We train baselines with $10^5$ steps with a batch size of 512. CSMPN uses a batch size of 16 in the training process.

## E.2 CMU MOTION CAPTURE

We use the same preprocessing as Huang et al. (2022). The processed dataset has 200 entries in the training set and 600 entries both in the validation set and test set. We manually lift the triangles formed by elbow, shoulder, and forearm nodes. Consequently, the edges and nodes that form the triangles are also lifted as simplices. Triangles formed by hip nodes, knee joint nodes, and heel nodes are also lifted to simplices. We also lift the rest of the nodes to zero-dimensional simplices and keep their connectivity intact. The architecture of CSMPN (200K) is kept similar to the Convex Hulls experiment. In Figure 7, we depict the initial, ground truth, and predicted positions by CSMPN.

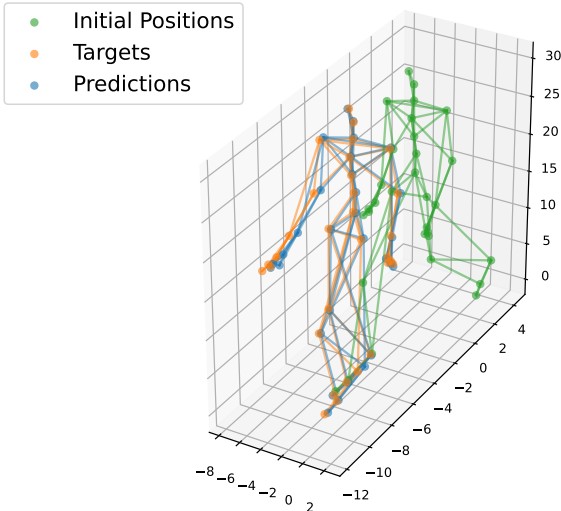

Figure 7: Example initial, ground truth target, and predicted positions of a CMU motion data sample.

The Adam optimizer with learning rate $5 \times 10^{-4}$ and weight decay $1 \times 10^{-5}$ was used for training. We train all models with $10^5$ steps with batch size $100$.

### E.3 MD17 ATOMIC MOTION PREDICTION

The same preprocessing method as Xu et al. (2023) is applied to the MD17 dataset. We sample and extract the positions of the heavy atoms from each of the molecules to form the final dataset. We use $k$-nearest neighbors to construct the Aspirin graph and fully connect the other molecules. We then *clique*-lift the graphs to form the simplicial complexes.

The training set and test set of each molecule have 5000 and 2000 instances, respectively. For Aspirin, we have 1303 validation instances. For the other molecules, we have 2000 instances to validate and tune the model performance.

We use a similar structure of CSMPN (300K) but with 5 message passing layers, each using Clifford group-equivariant networks with 32 hidden neurons as message and update functions. An Adam optimizer is used to train CSMPNs with learning rate $1 \times 10^{-3}$ and weight decay $1 \times 10^{-6}$. We train all models with $10^5$ steps, with each batch containing 100 instances.

### E.4 NBA PLAYERS TRAJECTORY PREDICTION

The same preprocessing steps from (Monti et al., 2020) are applied to obtain the final trajectory dataset. 8420 entries are used to train the CSMPN. We have 2806 entries in the validation test sets, respectively. All five players are connected to each other to form fully connected graphs. A clique complex lift is used to form the simplcial complexes from regular graphs. We use 4 simplicial message-passing layers with Clifford group-equivariant networks with 32 hidden neurons as message and update functions, yielding roughly 200K parameters. An Adam optimizer with learning rate $5 \times 10^{-3}$ is used to train CSMPN. We train all models with $10^5$ steps, with each batch containing 100 instances.

## F ALGORITHMS

In Algorithm 2, Algorithm 3, and Algorithm 4, we denote message passing, simplicial message passing, and *shared* simplicial message passing, respectively. Here, $\mathrm{Embed}$ and $\mathrm{Readout}$ are neural networks that embed input data and project the final node features to a single output, respectively. We also included graph and simplicial pooling aggregators. Considering that the *1-skeleton* of a

simplicial complex is a graph, and regular message passing only uses the upper adjacencies, it is clear that standard message passing is a special case of simplicial message passing. Further, we see that we can effectively share parameters in the shared setting while leveraging the simplicial complex by conditioning on the source and target simplices. Finally, by reducing the incoming message to be an aggregation of messages coming from varying simplex dimensions, we can use existing CUDA-optimized graph reduction operations already present in graph neural network libraries, e.g., Fey & Lenssen (2019).

---

**Algorithm 2** Standard Message Passing

---

**Require:** $G = (V, E), \forall v \in V : h_{\text{in}}^v, \phi^m, \phi^h$

$\quad h_0^v \leftarrow \text{Embed}(h_{\text{in}}^v)$
$\quad \textbf{for } \ell = 0, \ldots, L - 1 \textbf{ do}$
$\quad\quad \text{\# Message Passing}$
$\quad\quad m_\ell^v \leftarrow \text{Agg}_{w \in N(v)} \phi^m(h_\ell^v, h_\ell^w)$
$\quad\quad h_{\ell+1}^v \leftarrow \phi^h(h_\ell^v, m_\ell^v)$
$\quad \textbf{end for}$
$\quad h^G \leftarrow \text{Agg}_{v \in V} h_L^v$
$\quad h_{\text{out}} \leftarrow \text{Readout}(h^G)$
$\quad \textbf{return } h_{\text{out}}$

---

---

**Algorithm 3** Simplicial Message Passing

---

**Require:** $K, \forall \sigma \in K : h_{\text{in}}^\sigma, \phi^m, \phi^h$

$\quad h_0^\sigma \leftarrow \text{Embed}(h_{\text{in}}^\sigma)$
$\quad \textbf{for } \ell = 0, \ldots, L - 1 \textbf{ do}$
$\quad\quad \text{\# Message Passing}$
$\quad\quad m_\ell^B(\sigma) \leftarrow \text{Agg}_{\tau \in B(\sigma)} \phi_B^m(h_\ell^\sigma, h_\ell^\tau)$
$\quad\quad m_\ell^C(\sigma) \leftarrow \text{Agg}_{\tau \in C(\sigma)} \phi_C^m(h_\ell^\sigma, h_\ell^\tau)$
$\quad\quad m_\ell^{N_\uparrow}(\sigma) \leftarrow \text{Agg}_{\tau \in N_\uparrow(\sigma)} \phi_{N_\uparrow}^m(h_\ell^\sigma, h_\ell^\tau)$
$\quad\quad m_\ell^{N_\downarrow}(\sigma) \leftarrow \text{Agg}_{\tau \in N_\downarrow(\sigma)} \phi_{N_\downarrow}^m(h_\ell^\sigma, h_\ell^\tau)$
$\quad\quad h_{\ell+1}^\sigma \leftarrow \phi^h(h_\ell^\sigma, m_\ell^B(\sigma), m_\ell^C(\sigma), m_\ell^{N_\uparrow}(\sigma), m_\ell^{N_\downarrow}(\sigma))$
$\quad \textbf{end for}$
$\quad h^K \leftarrow \text{Agg}_{\sigma \in K} h_L^\sigma$
$\quad h_{\text{out}} \leftarrow \text{Readout}(h^K)$
$\quad \textbf{return } h_{\text{out}}$

---

---

**Algorithm 4** Shared Simplicial Message Passing

---

**Require:** $K, \forall \sigma \in K : h_{\text{in}}^\sigma, \phi^m, \phi^h$

$\quad h_0^\sigma \leftarrow \text{Embed}(h_{\text{in}}^\sigma)$
$\quad \textbf{for } \ell = 0, \ldots, L - 1 \textbf{ do}$
$\quad\quad \text{\# Message Passing}$
$\quad\quad m_\ell^\sigma \leftarrow \text{Agg}_{\substack{\tau \in B(\sigma) \\ \tau \in C(\sigma) \\ \tau \in N_\uparrow(\sigma) \\ \tau \in N_\downarrow(\sigma)}} \phi^m(h_\ell^\sigma, h_\ell^\tau, \dim \sigma, \dim \tau)$
$\quad\quad h_\sigma^{\ell+1} \leftarrow \phi^h(h_\sigma^\ell, m^\ell, \dim \sigma)$
$\quad \textbf{end for}$
$\quad h^K \leftarrow \text{Agg}_{\sigma \in K} h_L^\sigma$
$\quad h_{\text{out}} \leftarrow \text{Readout}(h^K)$
$\quad \textbf{return } h_{\text{out}}$

---

| Method | Dataset | Seconds/Step |
|---|---|---|
| EMPSNs | Aspirin | 0.04 |
| CEGNNs | Aspirin | 0.08 |
| CSMPNs (separated) | Aspirin | 0.37 |
| **CSMPNs (shared)** | Aspirin | 0.22 |
| EMPSNs | Benzene | 0.03 |
| CEGNNs | Benzene | 0.07 |
| CSMPNs (separated) | Benzene | 0.40 |
| **CSMPNs (shared)** | Benzene | 0.24 |
| EMPSNs | Ethanol | 0.03 |
| CEGNNs | Ethanol | 0.08 |
| CSMPNs (separated) | Ethanol | 0.29 |
| **CSMPNs (shared)** | Ethanol | 0.09 |
| EMPSNs | Malonaldehyde | 0.03 |
| CEGNNs | Malonaldehyde | 0.07 |
| CSMPNs (separated) | Malonaldehyde | 0.32 |
| **CSMPNs (shared)** | Malonaldehyde | 0.16 |

Table 5: inference time of Clifford, Shared and Separate message passing networks on MD17 atomic motion dataset.

## G    SHARED VS SEPARATE MESSAGE PASSING LAYERS

To provide a practical perspective on the shared message passing scheme, we present a table on the inference times of CSMPNs when applied to the MD17 atomic motion dataset. Averaged inference time is measured for a batch of 100 samples running on a GeForce RTX 3090 GPU. For a comprehensive understanding, we also include performance metrics of CEGNNs (Clifford Group Equivariant Graph Neural Networks) and CSMPNs with separate simplicial message passing networks, serving as comparative benchmarks.

