# OpenReview forum: "Clifford Group Equivariant Simplicial Message Passing Networks"
_ICLR.cc/2024/Conference — ICLR 2024 poster_

### Official Review · Reviewer_u7om · 2023-10-20

**Soundness:** 3 good
**Presentation:** 3 good
**Contribution:** 2 fair
**Rating:** 6
**Confidence:** 4

**Summary:**

- The authors propose a new architecture for geometric graphs that combines simplical message-passing, Clifford algebras, and equivariance.
- On a high level, their approach is based on linking the expansion in the number of vertices that occurs in simplices with the expansion in grades in a Clifford algebra.
- A given geometric graph is first lifted to simplices, then the outputs are computed with a message-passing algorithm between the simplices. Different from Bodnar et al (2021), the proposed algorithm uses Clifford algebras as representations, the geometric product to compute simplex features, and Clifford Group Equivariant NNs (Ruhe et al, 2023) to construct messages. These changes add a geometric inductive bias to the algorithm and make it equivariant.
- The algorithm is demonstrated on experiments ranging from simple toy data to motion capture data, molecular dynamics, and even NBA data.

**Strengths:**

- The idea is novel and interesting. I like how the authors link two previously disjointed ways of describing expansions to multiple objects.
- The experimental evaluation on a range of different problems highlights the versatility of the approach. The results demonstrate that the method works.
- The paper is well-written, the structure clear, and the figures of a high quality.

**Weaknesses:**

- The authors overstate the naturality of their method. While the approach feels intuitive at a quick glance, it mixes two different concepts: Simplices are a topological concept, while the geometric product is related to the geometric space that the graph is embedded in. This leads to a number of issues. Even in 3D space, one can construct 100-simplices, but the Clifford algebra only goes up to grade 4. While intuitively higher-order simplices encode more information, higher-order grades eventually become lower-dimensional. More generally, the geometric product is lossy: different simplices will be described by the same multivector. These all seem like undesirable properties for a simplex representation. I wish the authors would acknowledge and discuss these issues, which to me seem central to this paper.
- While I like the experiments, it would be useful to see the performance on more standard benchmarks of geometric deep learning. The authors acknowledge that they do not achieve SOTA results on some of these tasks. That's fine, but it would still be useful to see how the method performs compared to other generic equivariant methods.
- The paper's contributions are somewhat thin: the authors essentially plug together two existing ideas (simplical message passing and Clifford group equivariant networks). Of course, the same can be said about many other papers.

**Questions:**

- Could you comment on what I wrote in the first point in the Weaknesses section? It's well possible I missed something here.
- On page 2, what do you mean with "geometric products are meaningful in all such cases"?
- On page 7, you mention that you use an equal parameter count for CSMPN and the baselines. Is that really fair when compairing equivariant and non-equivariant baselines? Usually, equivariant methods have far less parameters for the same expressivity.

---

> ### Author Response · Authors · 2023-11-15
>
> We thank the reviewer for the highly valuable feedback. We address their concerns and questions in the following.
>
> 1. **Using the Clifford algebra in combination with simplicial message passing**
>     * *Is it reasonable to use Clifford algebra to represent 100-simplices in 3D space if Clifford algebra only has 4 subspaces?*  We agree that the naturality of the method might have been overstated, and have adjusted the language accordingly. We appreciate the reviewer's provision of this specific example, and we would like to use it as an opportunity to elaborate on the overarching concept of our paper. Our objective is to develop a model capable of comprehending both the topological and geometric aspects of geometric graphs. In 3D Euclidean space, Clifford algebra will indeed map 100-simplices to representations within four subspaces. However, note that the geometric features describable in 3D space are confined to positions (defined by 1 point), lengths (defined by 2 points), areas (defined by 3 points), and volumes (defined by 4 points). One might be interested in geometric features like angles between planes, but those are derivable from the positions, lengths, and areas. When the Clifford algebra is employed to represent a 100-simplex, it effectively condenses the simplex's geometric features into a combination of geometrically meaningful subspaces within the algebra. While a 100-simplex is defined topologically, the dimensional extent of its geometric features is limited by the space it occupies, which in this scenario is 3D space.
>     - *Can the Clifford algebra encode arbitrary higher-order simplices?*.
>      The reviewer correctly notes that Clifford algebra can constrain higher-order simplices to representations with limited grades based on the geometric space where the graphs are embedded. As we stated, we think that Clifford algebra is able to represent meaningful geometric features corresponding to the space in which the data lives. An alternative approach to capturing multiplicative geometric interactions within arbitrarily high-order simplices $\sigma^D$ could involve tensor products of the representations of vertices $\{v_i\}_{i=0}^D$. However, this method is computationally intensive due to the exponentially growing dimensions of tensor products and perhaps the associated calculations of irreps (e.g. Clebsch–Gordan coefficients). Although Clifford algebra might limit the expressiveness in representing complex higher-order interactions, one possible workaround is to employ additional hidden channels. These channels can represent higher-order simplices in larger functional spaces, potentially leading to more effective representations. In general, finding optimal machine-learning representations of higher-order objects is currently an active research area.
>      - *Will different simplices be described by the same multivector?*
>     We apologize for any lack of clarity in the main text regarding using Clifford algebra to learn simplicial features. For a detailed explanation of this process, particularly for higher-order simplices, we would like to refer to section 3.2 in the main text. Furthermore, we recommend going through a more detailed clarification, which we responded to reviewer q1Xe. We can infer from the Clifford simplicial feature embedding that the model is expected to learn unique simplices comprising distinct vertices features. This is because varying vertices typically have different geometric features (e.g. positions) within the geometric space, leading to distinct input features for the Clifford learnable layers. Consequently, this ensures that the output, namely the learned higher-order simplicial features, will be distinct for different simplices. We regret any ambiguity in the main text and hope this explanation adequately addresses your concerns.
>
>     * Finally, we would like to highlight that the Clifford algebra is not limited to three-dimensional space. Indeed, we carried out a supplementary experiment showing that including higher-dimensional simplices in this case improves the performance on the 5D convex hulls task. The simplex dimension is captured by the simplicial message passing framework, and our paper shows that fusing this with good geometric representations through the Clifford algebra is fruitful.
>
>     | Method                    | MSE (↓) |
>     |---------------------------|---------|
>     | CSMPNs (max dim = 2)      | 0.002   |
>     | CSMPNs (max dim = 3)      | 0.002   |
>     | CSMPNs (max dim = 4)      | 0.001   |
>
>     *Table 2: Convex Hull experiments with an increased maximal simplex dimension using CSMPNs*
>
> **Please see the next comment for the rest of our response**

---

> > ### Comment · Reviewer_u7om · 2023-11-20
> >
> > Thank you for the thorough response.
> >
> > I appreciate the discussion of the relation between topology and geometry and the promised changes to the writing – I believe that toning down the use of labels like "natural" and "meaningful" and expanding some discussions will improve the paper.
> >
> > Thanks for clarifying **how simplical features are constructed** – like another reviewer, I misunderstood the paper and was under the impression that you just use geometric products for the simplical embeddings. It now makes sense to me.
> >
> > As for the discussion about **comparisons with equal parameter counts**, I understand your choice, but I'm not entirely convinced that it is clearly better than comparing based on equal channels and layers. Of course, ideally one would tune all methods separately within a fixed compute budget, but that is expensive.
> >
> > Overall, while I still think that the paper is on the thin side and the empirical evaluation is not entirely convincing, it does make a worthwhile contribution, and I lean towards recommending acceptance. I have updated my score to reflect that.

---

> > > ### Author Response · Authors · 2023-11-21
> > >
> > > We wanted to again extend our sincere gratitude to the reviewer for their constructive suggestions and feedback. Also we appreciate the reviewer's positive reaction. We have checked and tried to modify the main text to make the Clifford simplicial feature clear and understandable.

---

> ### Author Response · Authors · 2023-11-15
>
> 2. **What do we mean by "geometric products are meaningful in all such cases"?** We realized and agreed that the description of simplicial feature initialization in the text is not accurate and clear. We updated this in the main text within spatial constraints and included an additional explanation here for completeness.
>     The intention of the sentence was that *whatever space or dimension* we work with, we can embed simplices with something multiplicative that is also geometrically meaningful through the geometric product. Take, for example, a 1-simplex $\sigma$ formed by vertices ${v_i, v_j}$; we can initialize this simplex using the geometric product of their respective Clifford features $h^{v_i}$ and $h^{v_j}$. If these features are derived from embedding the positions $p_i$ and $p_j$ of $v_i$ and $v_j$, the geometric product will result in a multivector comprising both grade 0 and grade 2 elements, namely a scalar and a bivector. In this multivector, the scalar component represents the similarity of vectors $p_i$ and $p_j$ relative to the origin. Conversely, the bivector component signifies the vector perpendicular to the plane spanned by $p_i$ and $p_j$. This geometric interpretation can similarly be applied to simplices of higher dimensions. However, as the reviewer noted, the scope of this geometric interpretation is bounded by the dimension of the Clifford algebra or, more specifically, by the geometric space in which the graphs are embedded.
>
> 3. **Model comparisons with an equal parameter count**
>     We appreciate the reviewer's insightful observation regarding the efficiency of equivariant models. It is indeed a recognized advantage that equivariant models can achieve comparable expressiveness with fewer parameters than their non-equivariant counterparts. This efficiency is one of the key benefits of using equivariant models.
>     However, in our study, we deliberately chose to compare the CSMPNs with equivariant baselines using an equal parameter count. Our intention was to provide a balanced framework for comparison, ensuring that any performance differences observed are not merely a result of varying parameter quantities.
>     Moreover, we included basic non-equivariant models, such as Message Passing Neural Networks, in our comparison to offer a more holistic view. We believe that this inclusive approach allows for a broader and more insightful evaluation, showcasing how CSMPNs perform relative to both equivariant and non-equivariant models. This comparative analysis aims to shed light on the practical implications and advantages of CSMPNs in diverse modeling scenarios.
>     We apologize if the rationale behind our methodological choices was not clearly conveyed in the main text, and we hope this response and updates to the text address the concerns raised.
> 4. **Other benchmarks in geometric deep learning and contributions.** Thank you for the suggestions! Preliminary experiments we conducted on QM9 (L. Ruddigkeit et al, 2012.) and n-Body system datasets (for n-Body dataset, we matched SEGNN (Brandstetter, J.et al (2021)) performance on initial tries) looked promising, but soon we figured out that the SOTA architectures were too optimized using domain knowledge and hence we decided to take a different route. It is not within our resource constraints to run these experiments during the rebuttal period.
>
>       Regarding the significance of contributions, integrating simplicial message passing with steerable GDL methods is rather unexplored territory. Our initial study has subtly suggested potential further avenues for combining geometry and topology within graph learning tasks, which we hope may offer some meaningful addition to this growing area of study. Furthermore, we would like to point out that we were presented with computational challenges, which we alleviated by suggesting strategies such as sharing simplicial message passing layers across different simplex orders.
>
> We want to reiterate our thanks to the reviewer for their insightful and valuable feedback. We hope that our responses above have sufficiently addressed any questions and concerns. We are ready and willing to provide further clarifications in subsequent iterations if needed. Finally, we hope the reviewer may reconsider and improve their scores in light of our responses.
>
> **Actions taken**: We have modified the main text to avoid ambiguity and tried to make the Clifford simplicial learnable embedding part easy to understand.

---

### Official Review · Reviewer_q1Xe · 2023-10-30

**Soundness:** 3 good
**Presentation:** 3 good
**Contribution:** 3 good
**Rating:** 6
**Confidence:** 3

**Summary:**

In this paper, the authors proposed a Graph Neural Networks method that combines both simplicity message passing and Clifford group equivariants neural networks. It is understood that this is an extension of the recent study of E(n) equivariant message passing simplicity networks, which has the limitation of manual initialization and restriction of updating methods. Numerical experiments show that the proposed method outperform existing methods in most cases.

**Strengths:**

As I have summarized above, the min contribution of this paper is to propose an alternative message passing network to the existing EMPSN method, and eliminate two main limitations of EMPSN.

Although neither utilizing Clifford group in neural networks  nor simplicial message  passing is new, the authors combine them to achieve an advancement in GNN.

**Weaknesses:**

In the key technical session 3.2, embedding simiplicial data to Clifford algebra, the embedding of the simplices is not fully and clearly described, thus renders a full understanding of the proposed method difficult. The authors did point to Figure 1 for depiction of the method, but that is open for different interpretations.

While numerical results indeed show that new method, CSMPN, achieves the best performance in most experiments, the improvement over existing methods can not be characterized as significant, thus not leading to a convincing argument for its usage.

**Questions:**

Maybe the authors can explain by "learn" the embedding through Clifford group-equivariant layers in Sec. 3.2.

---

> ### Author Response · Authors · 2023-11-15
>
> We thank the reviewer for the actionable and valuable feedback. We address their concerns and questions in the following.
> ​​
> 1.  **Clarification regarding the learned simplicial embeddings** We have incorporated the reviewer's feedback and clarified Section 3.2 within space constraints. For completeness, we detail the procedure here.
>  Before describing the details of learnable Clifford simplicial embeddings, we would like to depict its high-level procedure for better and generalized understanding.  Suppose that we have a graph G = {V, E}. To generate the Clifford simplicial embeddings, we do the following:
>     - For each vertex $v_i \in V$, we embed the node features, e.g., positions, velocities, and atomic charges in Clifford space, to obtain Clifford features $h^{v_i} \in Cl(V, q)$.
>     - Use lifting methods to lift the regular graph $G$ to a simplicial complex $K$, with $\sigma_i$ denoted as the i-th simplex in the simplicial complex $K$.
>     - For each simplex $\sigma_i \in K$, we use the features of set of vertices $\{v_i, v_j, \dots \}$ that compose the simplex $\sigma_i$ to generate the (learnable) Clifford features $h^{\sigma_i}$ for simplex $\sigma_i$.
>
>
>
>     We now consider a simplicial complex $K$ lifted from $V$. One can consider two methods to extract simplicial embeddings from this complex.
>
>
>
>     - *Non-Parametric Simplicial Clifford Features*
>     We utilize a non-parameterized approach to acquire a Clifford feature $h^\sigma$ for each simplex $\sigma \in K$. For individual vertices (i.e. vertex) $\{ v_i \} \in K$, one can directly assign $h^{\{ v_i \}} := h^{v_i}$, embedding the node attributes into Clifford space using Clifford algebra. For 1-simplex, i.e. edges, the Clifford feature can be the geometric product of the individual vertex features, denoted as $h^{\{ v_i, v_j \}} := h^{v_i} h^{v_j}$. This process extends to higher-dimensional simplices, like triangles $\{ v_i, v_j, v_k \} \in K$,  where we use the geometric product of the three Clifford features, i.e., ${ h^{\{ v_i, v_j, v_k \}} := h^{v_i} h^{v_j} h^{v_k}}$. Alternative methods for initializing simplicial features include mutual differences or averaged features. In our implementation, however, we opt to learn simplicial features using Clifford equivariant layers.
>     - *Learned simplicial Clifford features*
>     Similarly, our aim is to obtain a learned Clifford features $h^\sigma$ for each simplex $\sigma \in K$. For the singletons $\{ v_i \} \in K$,  we use the Clifford-space embedded representation $h^{\{ v_i \}} := h^{v_i}$ as an input to a Clifford feature learning layer, e.g. a linear Clifford equvariant layer, to generate learnable features. For the edges $\{ v_i, v_j \} \in K$, we can use the stacked Clifford-space representations $(h^{v_i}, h^{v_j})$ as an input to a bilinear Clifford equivariant layer (Ruhe, D., Brandstetter, J., \& Forré, P. (2023). Clifford group equivariant neural networks.), which contains the learnable geometric product module. To make the resulted simplicial Clifford features permutation invariant to the node ordering by aggregating permutations of geometric products. A more intricate way of doing so would be to pass messages between a $k$-simplex's constituent vertices $k$ times. The (aggregated) readout is then sent to the $k$-simplex as its initialized feature. In practice we found aggregating permutations of geometric products was sufficient. This process can similarly be extended to higher-order simplices while we do restrict the number of layers to match the dimension of the simplex with the grade of the the Clifford feature.
> ​
> 2. **Nonsignificant improvements over baselines.**  We would like to highlight the consistency of the improvement of CSMPNs compared to baseline models across multiple datasets. This trend suggests an effective fusion of topological and geometric concepts within our model. These advancements suggest that CSMPNs are a strong candidate for handling datasets with complex topological and geometric characteristics. Furthermore, it paves the way for further research into integrating geometric and topological elements. We also would like to mention that one challenge encountered in our research with simplicial message passing is its computational complexity. We think that sharing message-passing layers across different simplex orders is a viable strategy to reduce the computational demand, and we are hopeful about further improvements in this area.
>
> Once more, we extend our sincere thanks for the reviewer's input and appreciate their broadly positive evaluation. We welcome any further questions or points for discussion and are keen to engage in additional conversations. Finally, we hope our response has addressed the concerns sufficiently and the reviewer will consider promoting their scores​.
>
> **Actions taken**: We have clarified Section 3.2.

---

### Official Review · Reviewer_atRt · 2023-10-31

**Soundness:** 3 good
**Presentation:** 3 good
**Contribution:** 3 good
**Rating:** 6
**Confidence:** 3

**Summary:**

The paper introduces Clifford Group Equivariant Simplicial Message Passing Networks (CSMPNs), a method designed for steerable $E(n)$-equivariant message passing on simplicial complexes. The approach combines the expressivity of Clifford group-equivariant layers with simplicial message passing, which is more expressive than regular graph message passing. To implement it, the paper discusses methods for lifting a set of points to a simplicial complex and embedding simplicial data in the Clifford algebra, where various types of lifts like Vietoris-Rips and manual lifts are considered. The results and evaluations of the CSMPNs are across various domains showing the effectiveness of this method.

**Strengths:**

1. The paper introduces a combination of Clifford group-equivariant layers and simplicial message passing, providing a strong theoretical foundation for the method. The use of Clifford algebras to represent geometric features with simplices is natural and mathematically sound.
2. The model is designed to be applicable across various domains, from geometry to molecular dynamics. This broad applicability is a strong point, especially for a mathematical audience interested in universal structures.

**Weaknesses:**

1. I would like to see the detailed inference time of CSMPN compared to others, e.g., in the MD17 atomic motion dataset. I believe the time complexity of CSMPN is highly relevant to the number of simplices, and I would like to know if this is an issue in the implementation.
2. As the authors mention experiments on MD17 and QM9 are beyond the scope of their research; I hope the authors give a discussion or possible directions on how the manual lift could be implemented for small and medium-sized molecules to strengthen the paper's applicability, not just using H2O as an example.

**Questions:**

See weaknesses.

---

> ### Author Response · Authors · 2023-11-15
>
> We thank the reviewer for the actionable and valuable feedback. We address their concerns and questions in the following.
> ​
> 1.  **Inference time of CSMPNs.**      CSMPNs are adept at processing simplicial complexes, resulting in higher computational complexities. To address this, we have implemented several design choices to parallelize training and inference phases. What resulted is what we term *Shared Simplicial Message Passing*.  The layers facilitate concurrent messages passing across simplices of varying dimensions, converting complex structures into regular graphs, which can be efficiently processed by well-written libraries like PyTorch Geometric. This method decreases network parameters and enhances parallel processing, thus speeding up both the training and inference phases.
>
>     To provide a practical perspective on this advancement, we present a table (also included in the appendix) on the inference times of CSMPNs when applied to the MD17 atomic motion dataset. Averaged inference time is measured for a batch of 100 samples running on a GeForce RTX 3090 GPU. For a comprehensive understanding, we also include the inference time of EMPSNs (Equivariant Message Passing Simplicial Neural Networks), CEGNNs (Clifford Group Equivariant Graph Neural Networks), and CSMPNs with *separated simplicial message passing* layers (which is the typical approach used by e.g. Bodnar et al. (2021). We want to highlight that EMPSNs scalarize geometric features, in contrast to steerable methods. This inherently reduces computational complexity.  CSMPNs with *shared simplicial message passing* layers surpass the ones with *separate simplicial message passing* layers by a large margin.
>     Research into further reducing the time/computation complexity of simplicial message passing framework in general without losing expressiveness provides an exciting future work direction.
>     | Method                | Dataset        | Seconds/Step |
>     |-----------------------|----------------|-------------:|
>     | EMPSNs                | Aspirin        |        0.04  |
>     | CEGNNs                | Aspirin        |        0.08  |
>     | CSMPNs (separated)    | Aspirin        |        0.37  |
>     | CSMPNs (shared)       | Aspirin        |        0.22  |
>     |                       |                |              |
>     | EMPSNs                | Benzene        |        0.03  |
>     | CEGNNs                | Benzene        |        0.07  |
>     | CSMPNs (separated)    | Benzene        |        0.40  |
>     | CSMPNs (shared)       | Benzene        |        0.24  |
>     |                       |                |              |
>     | EMPSNs                | Ethanol        |        0.03  |
>     | CEGNNs                | Ethanol        |        0.08  |
>     | CSMPNs (separated)    | Ethanol        |        0.29  |
>     | CSMPNs (shared)       | Ethanol        |        0.09  |
>     |                       |                |              |
>     | EMPSNs                | Malonaldehyde  |        0.03  |
>     | CEGNNs                | Malonaldehyde  |        0.07  |
>     | CSMPNs (separated)    | Malonaldehyde  |        0.32  |
>     | CSMPNs (shared)       | Malonaldehyde  |        0.16  |
> ​
>     *Table 1: Inference time comparison on MD17 atomic motion dataset*
> ​
>
>
> **​Please see the next comment for the rest of our response.**

---

> ### Author Response · Authors · 2023-11-15
>
> 2. **Discussion and direction of manual lift.**  Thanks for the feedback! We now include promising directions for testing manual lifts in the appendix (section D), also detailed here for completeness. There are various methods to transform molecular graphs into simplicial complexes. To use a manual lift, graphs can be initiated with the Vietoris-Rips lift to form simplicial complexes. Subsequently, one can impose manually set thresholds for different dimensional simplices (like edge lengths for 1-simplices, areas of triangles for 2-simplices, and volumes of tetrahedra for 3-simplices). A group of (geometrically) close nodes form higher-order chemical structures, such as functional groups (e.g., aldehydes, ketones, nitrates, etc). For example, if we want to predict some properties of (amino) acids, including a simplex corresponding with a hydrophilic group such as carboxyl could be useful. Additionally, since molecular graphs often contain carbon rings, it's logical to break these rings into 2-dimensional or higher-order simplicial complexes to better represent the rings in a topological sense. Moreover, higher-order simplices can be manually constructed based on expert knowledge to capture the specific chemical and physical properties of the molecules. For instance, in biochemical structures like proteins or DNA, certain arrangements of atoms or residues have specific functional implications. By manually constructing higher-order simplices, one can incorporate these domain-specific insights into the model. This approach allows for the explicit representation of essential molecular structures, such as hydrogen bonding patterns and aromatic systems, which are critical for understanding molecular behavior and interactions.
>
>
>
>
> We would like to thank again the reviewer for their insightful feedback, and we are grateful for their overall positive assessment. We have aimed to clear up any uncertainties and respond to the issues mentioned in the feedback. Should there be any more questions or concerns, we are fully prepared for further discussions. Lastly, we are hopeful that the reviewer might consider enhancing their scores based on our responses.
> ​
>
>
>
> **Actions taken**: We have added a few sentences to highlight the shared message passing scheme in the paper, included a table with inference times, and updated the text to clarify manual lift specifically related to molecules in a more detailed way.

---

### Author Response · Authors · 2023-11-15

We thank the reviewers for their detailed evaluation of our paper. The positive reactions and helpful suggestions have greatly improved our research.
In revising our paper, we have carefully considered each reviewer's feedback by providing clarifications, incorporating suggestions, and answering remaining questions. We also clearly indicated which parts of the paper were updated based on the reviewers' comments.

Particularly:
- We added a section in the supplementary material to highlight the inference time comparison.
- We added a paragraph in the appendix to describe directions one can use for manually lifting molecular / biology-related graphs
- We rephrased Section 3.2 to make it more explicit and understandable.
- We rephrased the description for simplicial feature initialization using the geometric product.


To us, Clifford Simplicial Message Passing Networks (CSMPNs) provide an exciting direction for combining steerable geometric methods with topology. We have analyzed how involving the Clifford algebra can lead to more expressive geometric features when combined with message passing on simplicial complexes, leading to better performance on various datasets. We are happy to see that the reviewers see the merit of our work.

---

### Meta-Review · Area_Chair_Z23X · 2023-12-12

**Metareview:**

Summary: The article introduces Clifford group equivariant simplicial message passing networks.

Strengths: Referees found the article provides an interesting combination of known elements in a way that addresses certain limitations of existing methods, with strong theoretical foundations. Experiments indicate the method can outperform other methods albeit not significantly in a variety of tasks.

Weaknesses: Concerns included inference time comparisons, which were addressed in the responses by providing inference times along with design choices to parallelise training and inference. Authors still indicate that further reducing the time complexity remains a direction of future work.

At the end of the discussion period, three referees were marginally positive about this article. Overall, I share the view of one of the reviewers, that even though the article could still be stronger in a few regards, it makes a worthwhile contribution.Therefore, I am recommending accept. I will ask that the authors carefully consider the reviewers feedback when preparing the final version of the document and any improvements not conducted during the rebuttal period.

**Justification For Why Not Higher Score:**

Novelty and empirical benefits are moderate.

**Justification For Why Not Lower Score:**

Generally positive evaluation with strengths outweighing weaknesses.

---

### Decision · Program_Chairs · 2024-01-16

Accept (poster)